# Study on the variable length simple pendulum oscillation based on the relative mode transfer method

Yang Yu[1], Jing Ma[1], Xiangli Shi[2], Jiabin Wu[1], Shouyu Cai[1], Zilin Li[1,3], Wei Wang[1,3], Hongtao Wei[1,3]*, Ronghan Wei[1,3,4]*

1 School of Mechanics and Safety Engineering, Zhengzhou University, Zhengzhou, China, 2 School of Computer and Artificial Intelligence, Zhengzhou University, Zhengzhou, China, 3 Engineering Technology Research Center of Henan Province for MEMS Manufacturing and Applications, Zhengzhou University, Zhengzhou, China, 4 Institute of Intelligent Sensing, Zhengzhou University, Zhengzhou, China

☯ These authors contributed equally to this work.
* Weihtwei@zzu.edu.cn (HW); Weiprofwei@zzuedu.cn (RW)

**Data Availability Statement:** All relevant data are within the manuscript and its Supporting Information files.

**Funding:** The study was supported by Key Scientific Research Projects of Universities in

## Abstract

In this study, we employed the principle of Relative Mode Transfer Method (RMTM) to establish a model for a single pendulum subjected to sudden changes in its length. An experimental platform for image processing was constructed to accurately track the position of a moving ball, enabling experimental verification of the pendulum's motion under specific operating conditions. The experimental data demonstrated excellent agreement with simulated numerical results, validating the effectiveness of the proposed methodology. Furthermore, we performed simulations of a double obstacle pendulum system, investigating the effects of different parameters, including obstacle pin positions, quantities, and initial release angles, on the pendulum's motion through numerical simulations. This research provides novel insights into addressing the challenges associated with abrupt and continuous changes in pendulum length.

## Introduction

The pendulum model is a classical paradigm in physics, holding notable theoretical and practical importance [1–3]. The pendulum kind of oscillations with variable length is among the classical problems of mechanics. In engineering, various critical domains rely on mathematical models to elucidate the oscillatory behavior of pendulums with variable lengths. For example, the swing of the lifting system when cranes hoist cargo [4–6], seismic isolation systems for bridge foundations [7], and the control of horizontal vibration in high-rise buildings [8]. A pendulum with periodically changing length is also considered as a simple model of a child's swing. Belyakov [9] investigated the dynamical behavior of this model, deriving asymptotic expressions for the boundaries of unstable domains. He employed numerical methods to study the chaotic motion of a pendulum influenced by variations in the problem parameters. Seyranian [10] investigated the intricate (chaotic) behavior of a pendulum with length varying according to harmonic law. Wright [11] considered two forced dissipative pendulum systems,

Henan Province, Grant No. 21A130003 Dr. Hongtao Wei, Songshan Laboratory Project, Grant No.221100211000-01 Dr. Hongtao Wei, and National Natural Science Foundation of China, Grant No. 12202400 Dr. Wei Wang.

**Competing interests:** The authors have declared that no competing interests exist.

the pendulum with vertically oscillating support and the pendulum with periodically varying length, with a view to draw comparisons between their behaviour. Anderle [12] investigated the vibrational damping issue of a periodically length-changing pendulum and employed Lyapunov based adaptive estimation of the viscous friction at the string pivot to improve the controller performance.

Some investigations have engaged in both analytical solution and numerical simulation of nonlinear single pendulum dynamics. Pinsky [13] qualitatively analyzed periodically driven pendulum systems and demonstrated the existence of two periodic solutions. Through rigorous qualitative investigations into the equation governing the oscillations of a periodically length-changing pendulum, Zevin [14] obtained results that hold true for variations in pendulum length under most circumstances. Johannessen [15] introduced a method exclusively employing elementary functions to approximate the solution of the differential equation describing the motion of a large-angle initial velocity pendulum. Yang [16] employed a homotopy analysis method to explore approximate solutions. Fernandez-Guasti [17] provided an exact solution for a pendulum with uniformly changing length. Big-Alabo [18] derived an approximate periodic solution for the non-harmonic or non-sinusoidal response of a pendulum during moderate to large amplitude oscillations. They utilized an enhanced continuous piecewise linearization technique to deduce an approximate solution of the pendulum, enabling highly accurate determination of oscillations across the entire potential amplitude range. Wang [19] adopted the method of numerical analysis to establish a simulation analysis model to analyze the motion pattern of the pendulum when the relative speed and direction of the pendulum and the damper are different and the damping force of the pendulum is different. Rahayu [20] utilized the Mathematica software to numerically simulate the motion of a nonlinear pendulum driven by damping, investigating the chaotic behavior of the system. Clifford [21] tackled the dynamic issue of a parametric pendulum exposed to harmonic excitation on a rotating track. They carried out an inquiry utilizing both analytical and numerical analysis approaches to examine the dynamic responses across diverse parameters and initial conditions.

In summary, the previous studies on the oscillations of variable-length pendulums predominantly focused on models with periodically changing pendulum lengths. Further investigation is needed for pendulum vibration systems with more significant nonlinearities, particularly those involving abrupt changes in pendulum length.

This study establishes a model for a pendulum system with abrupt changes in pendulum length using the principle of the relative mode transfer method (RMTM). It designs simulations and experimental validations to demonstrate the accuracy of the approach. Computational examples are presented for a double-stopper pendulum system with varying parameters, yielding time-domain responses, frequency spectrum analyses, and phase-space diagrams. The RMTM is a novel approach derived from the classical Mode Transfer Method (MTM) techniques, emphasizing the concept of "relative." Specifically, during a mode transition, the previous mode "state" is recorded, and the post-transition "state" is treated as an iterative motion relative to the preceding state [22]. Building on this idea [23], it not only facilitates the study of complex structural impact vibration but also addresses vibration problems with changing boundary conditions that cannot be modeled using external forces. Examples include the vibration of a cantilever beam with changing length and a pendulum with varying rope length. Both of these systems fall under the category of variable boundary condition problems, yet they cannot be simplified using the principle of force integration to convert changing boundary conditions into external forces in the system equations. Consequently, they can only be addressed using the RMTM. This article presents novel perspectives for the study of analogous nonlinear problems.

## Model of oscillations in a variable-length pendulum

The equation of motion for a damped pendulum system with large angular displacement is given Eq (1)

$$\ddot{\theta}(t) + 2\omega\xi\dot{\theta}(t) + \omega^2\sin\theta(t) = 0 \tag{1}$$

Eq (1) is a nonlinear differential equation, typically not easily solved directly. In practical situations, people may employ numerical methods (such as Euler's method, Runge-Kutta method, etc.) to simulate the behavior of a damped pendulum vibration described by this equation. Where $\theta(t)$ represents the angular displacement of the pendulum, $\xi$ representing the damping coefficient, $g = 9.8m/s^2$ indicating the acceleration due to gravity, and $l$ representing the length of the pendulum.

Fig 1 shows schematic illustration of a single pendulum with two stops, where the angular displacement is delineated along the vertical axis, with the positive direction on the right and the negative direction on the left. At time $t_0$ the pendulum is released from a specific initial angle with an initial velocity of 0. Its angular displacement is represented by $\theta(t)$. When the time is $t_1$, the line of the pendulum is obstructed, and its angular displacement is expressed as:

$$\theta(t) = \theta(t_1) + \theta(t) \tag{2}$$

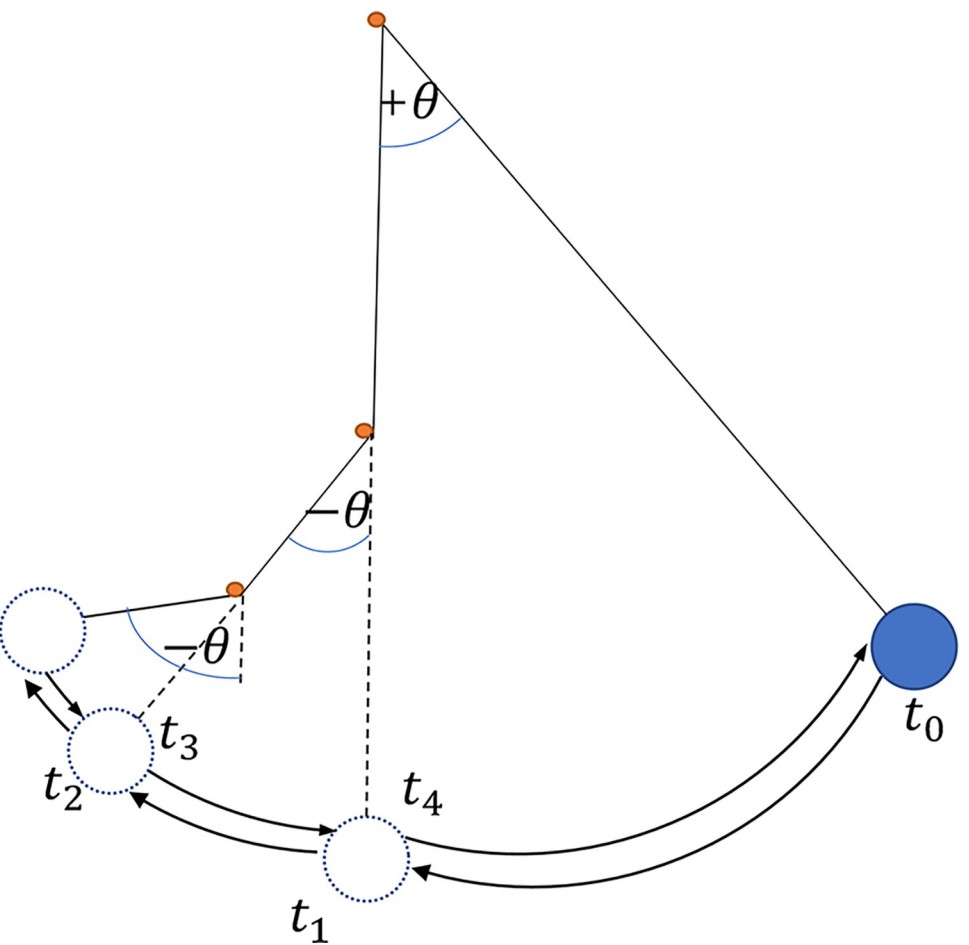

**Fig 1. Variable-length pendulum oscillation model.**

When reaching the time $t_2$, the pendulum is influenced by the second stop, resulting in an angular displacement of

$$\theta(t) = \theta(t_1) + \bar{\theta}_1(t_2) + \bar{\theta}_2(t) \tag{3}$$

When contact happens at the stop, by applying the concept of the RMTM, the initial values of angular displacement for each condition are set to be zero: $\bar{\theta}(t_1{}^+) = 0$ $\bar{\theta}(t_2{}^+) = 0$. The formula for angular velocity transformation during collision is $\dot{\bar{\theta}} = \bar{l}l\dot{\theta}$. $l, \theta$ represents the length and angular displacement of the pendulum when there is no obstruction. $\bar{l}, \bar{\theta}$ represents the length and angular displacement of the pendulum when obstructed. When obstructed, the angular displacement is measured from the obstruction point, with the left side considered positive and the right side considered negative. $\dot{\theta}$ represents the derivative with respect to time. Substituting Eq (3) into Eq(1), the oscillation equation for the single pendulum subjected to resistance is derived as

$$\ddot{\theta}(t) + 2\bar{\omega}\xi\dot{\theta}(t) + \bar{\omega}^2\sin[\theta(t_1) + \bar{\theta}_1(t_2) + \bar{\theta}_2(t)] = 0 \tag{4}$$

In this equation, $\bar{\omega}$ represents the natural frequency of the pendulum after encountering the stop, $\bar{\omega} = \sqrt{\frac{g}{\bar{l}}}$. At $t_4$, the pendulum returns to its initial rest length, which is not affected by the stop, until the velocity becomes zero near the release point. By extending the problem to a system with an arbitrary number of stops, for an arbitrary time $t_i$, where $t_i$ denotes the moments of changes in system boundary conditions, the angular displacement of the pendulum system can be written as:

$$\theta(t) = \theta(t_i) + \sum_{j=1}^{i-1}\theta(t_j) \tag{5}$$

By substituting Eq (5) into Eq (1), one can get:

$$\ddot{\theta}(t) + 2\omega_i\xi\dot{\theta}(t) + \omega_i{}^2\sin[\theta(t_i) + \sum_{j=1}^{i-1}\theta(t_j)] = 0 \tag{6}$$

During the collision, setting the initial angular displacements of each condition to zero, the transfer of angular velocity is given by

$$\dot{\theta}(t_{i^+}) = \frac{l_{i-1}}{l_i}\dot{\theta}(t_{i^-}) \tag{7}$$

In the above equation, $l_i$ represents the pendulum length during the time interval $t_i \sim t_{i+1}$ for the given condition. Eq (6) provides the differential equation for a damped large-angle pendulum with any number of stops. This equation lacks a strict analytical solution and can be numerically solved using the 4th-order Runge-Kutta method. Based on the RMTM, each condition is viewed as an individual motion state. At each transition point, the displacement and velocity values of the current moment are recorded. In the subsequent condition, the initial displacement is set to zero while the initial velocity is set to the final velocity of the previous condition. The ball continues to move according to the new system equation. Within this framework, the transition cycle continues. Each transition only alters the form of the system equation and passes parameters to the new equation.

## Numerical simulation and experimental comparison

In this chapter, MATLAB software is employed for the numerical simulation of the model. High-speed cameras are used to capture images of a single pendulum model, a single pendulum model with a single obstruction, and a double-obstructed single pendulum model. Motion parameters are obtained through image capture. The application of the concept of relative mode transformation to the reliability and applicability of the pendulum length mutation system is validated. This, in turn, provides a basis and methodology for the modeling and application of nonlinear pendulum systems.

**Experimental model.** The self-built experimental platform is shown in Fig 2 below. A coordinate board is fixed in the vertical plane. A blue metallic pendulum ball is used, and a signal-starting retroreflector is positioned to the right of the background board. A high-speed camera is placed approximately 0.3m in front of the pendulum, and its height is adjusted using a tripod. During the experimental process, a high-speed camera (CCD) is utilized to capture images of the pendulum system with obstruction, with a capture frame rate of 31 frames per second. To ensure the accuracy of the experimental results and minimize the impact of errors, a laser light source is initially used to illuminate the starting position. When the pendulum is set in motion, the laser reflection mirror reflects the signal, triggering the high-speed camera to start capturing. Fig 2 shows the high-speed imaging system for the obstructed pendulum.

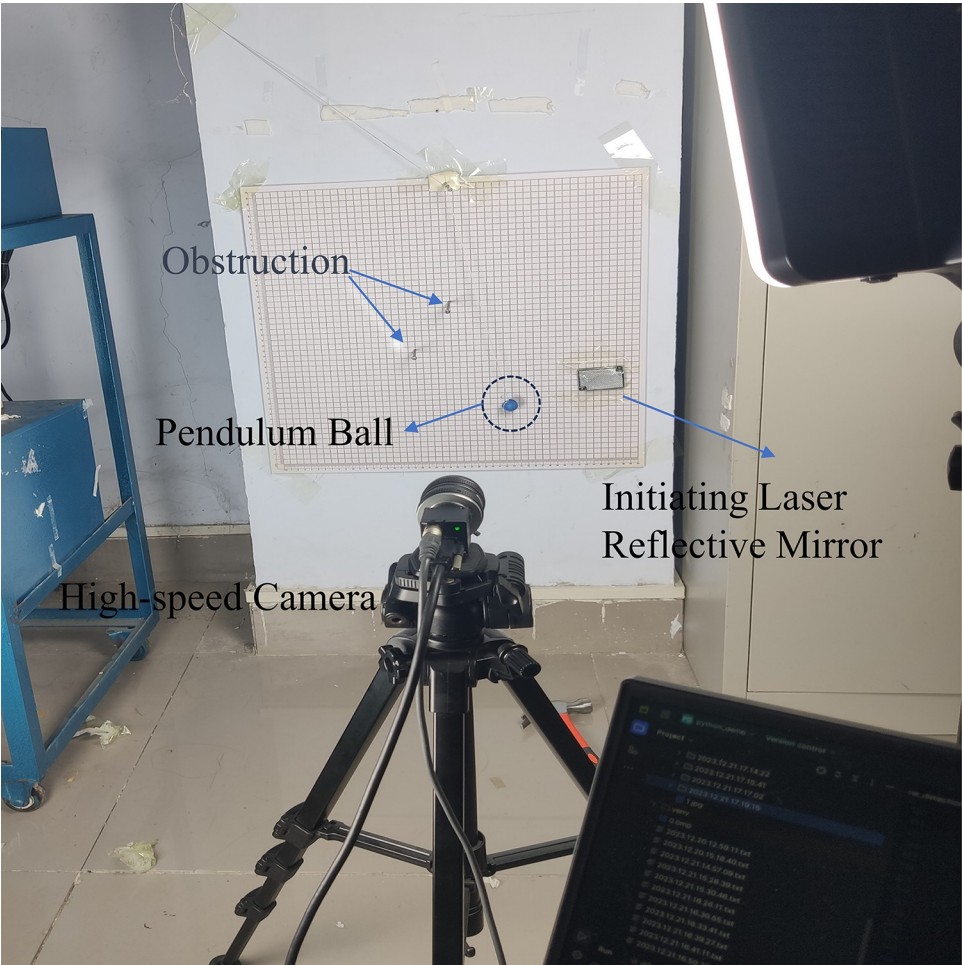

**Fig 2. High-speed camera system for pendulum with obstruction.**

Image processing of the captured photos from the camera mainly focuses on the detection and feedback of the pendulum's position coordinates in the images.

1. The images are subjected to simple processing, including grayscale conversion and median blur operations, to enhance detection accuracy. Converting BGR images to grayscale reduces the impact of noise in image processing.

2. The Canny edge detection algorithm is applied to perform edge detection on the images. Its primary purpose is to extract useful structural information and significantly reduce the amount of data to be processed, providing a small and precise data processing set for subsequent Hough circle detection.

3. The Hough circle detection algorithm is employed to record the feedback of the circular center positions of the balls in the images. Both algorithms can be implemented using built-in functions in OpenCV. After determining the pixel coordinates of the pendulum ball's center using the aforementioned methods, the world coordinates of the ball's center can be obtained through the transformation matrix M. Finally, a simple conversion yields the angle θ between the ball and the vertical direction.

This paper involves the development of a program to automatically process photos captured by a high-speed camera. The photos are recognized as pixel coordinates and then transformed into world coordinates and angles. In multiple sets of pixel coordinate recognition photos, four images of a pendulum without obstruction are selected, as shown in Fig 3, illustrating the process of undamped pendulum movement from the starting point to the maximum displacement on the left. The experiment records the motion of the pendulum through multiple cycles. The pixel coordinates of a double-obstructed pendulum, shown in Fig 4 from (a) to (d), depict the process of touching both obstructions, touching one obstruction, and not touching any obstruction, demonstrating the motion of the double-obstructed pendulum.

**Experimental conditions.** Three sets of experiments were conducted: Condition One, with no obstruction, the pendulum was released from the initial angle of 30˚ without any initial velocity; Condition Two, with one obstruction, the obstructed position is as shown in Table 1. When the pendulum string touches the obstruction, the pendulum length undergoes a sudden mutation, oscillating between obstructed and unobstructed states. Condition Three, with an additional second obstruction, positioned as indicated in the table. All other conditions remained constant. Repeat the test multiple times for each group of working conditions to ensure the accuracy of the experiment. The picture of simple pendulum angular displacement and time in working condition 1 is shown in S1 Fig. The picture of the angular displacement and time of the pendulum with an obstruction in the second operating condition is shown in S2 Fig. The picture of the angular displacement and time of the simple pendulum with an obstruction in the third operating condition is shown in S3 Fig.

**Comparison of results.** Experimental and simulation studies were conducted for the scenarios of a pendulum without obstruction, with one obstruction, and with two obstructions, denoted as Condition One, Condition Two, and Condition Three, respectively. The results were compared, as shown in Figs 5–7. The simulation utilized the fourth-order Runge-Kutta method to solve the system of differential equations, with a simulation time step of $\Delta t = 1 \times 10^{-4}$. The blue scattered points in the figures represent experimental data, and the red curves represent simulation data. In Conditions Two and Three, the blue dashed lines indicate the angles of the obstructions, and the angle displacement of the pendulum line is significantly abrupt when the obstruction occurs. The two sets of data closely match, with errors within an acceptable range. The results indicate that the simulation can accurately predict the motion of the large-angle damped pendulum and the large-angle damped pendulum with one or two

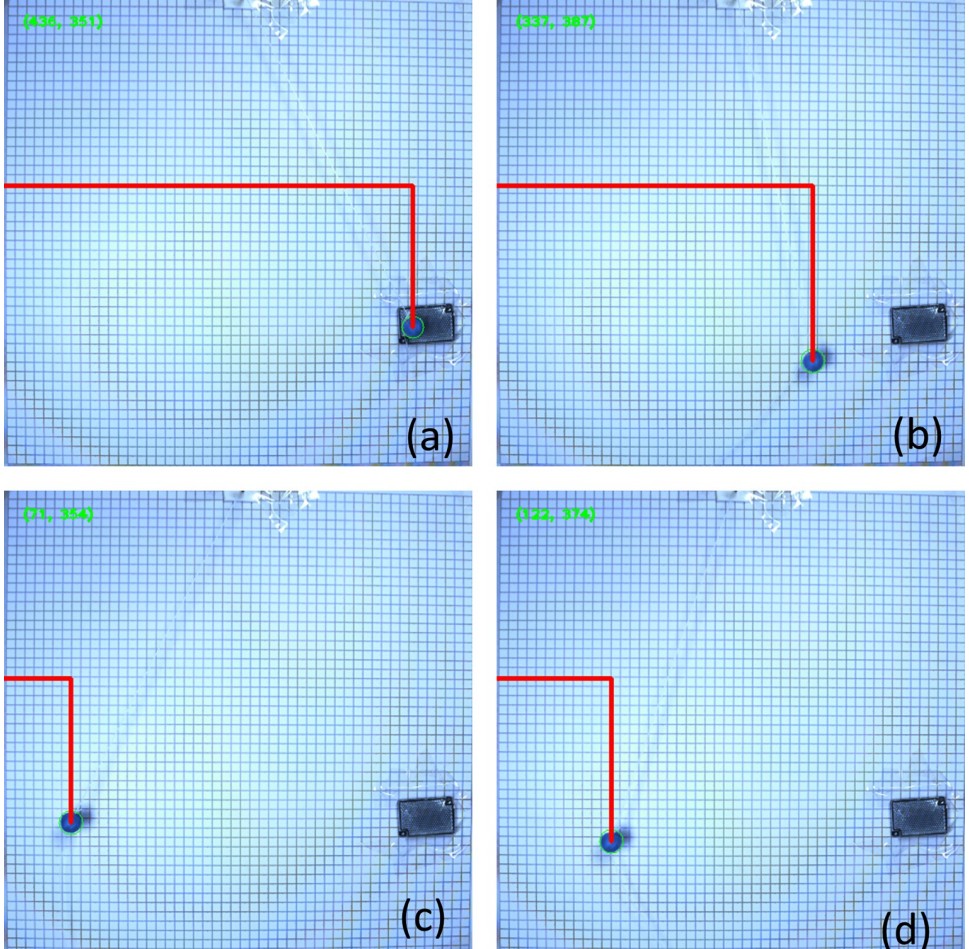

**Fig 3. Pixel coordinates recognition diagram for large-angle damped pendulum without obstruction.**

obstructions. This demonstrates that the model established through the Relative Mode Transformation Method (RMTM) can accurately solve the motion characteristics of the pendulum with significant angular displacement and changing pendulum length due to obstruction. The findings validate the applicability of this method in solving complex nonlinear vibration problems.

## Case study

In order to investigate the influence of nonlinear boundaries on the dynamics of a pendulum, numerical simulations were conducted to solve for the angular displacement response of the system. A simulation time step of $\Delta t = 1 \times 10^{-4}$ was employed, spanning a time period of 20 seconds. A comparison was conducted between scenarios featuring obstacles and those without, considering various initial release angles for an identical pendulum. Furthermore, the impact of different obstacle positions on the temporal response of the pendulum was investigated.

Furthermore, the Fast Fourier Transform (FFT) analysis was performed on the motion of the pendulum to conduct a spectral analysis of its oscillatory behavior. Phase portraits were generated to analyze the dynamics of the double-pendulum system under various conditions, considering the different obstacle configurations.

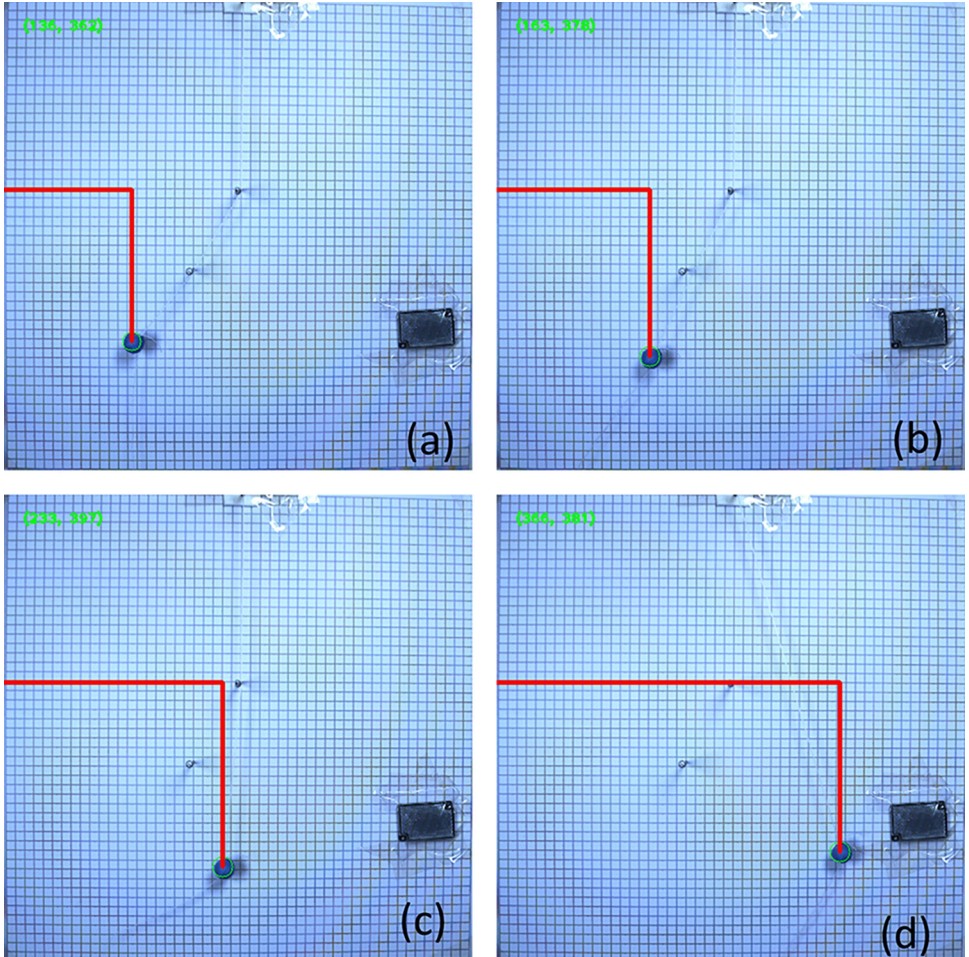

**Fig 4. Pixel coordinates recognition diagram for large-angle damped pendulum with double obstructions.**

**Swing and spectral analysis.** Under identical conditions, a numerical analysis was performed on both a double-pendulum system with obstacles and a single-pendulum system without obstacles. The parameters of the pendulum are listed in Table 2. A comparative study of the swing curves of the pendulum bob is presented in Fig 4A. Additionally, the frequency domain responses obtained through the FFT analysis are depicted in Fig 4B.

The motion state of the pendulum bob in the double-obstacle configuration is categorized into three operational conditions: Condition 1, where the pendulum swings without obstruction and has a length of

$l_1$; Condition 2, where the pendulum length becomes $l_2$ following the impact of the first obstacle; and Condition 3, where the pendulum length changes to $l_3$ after encountering the second obstacle. The initial angle is set to 34.9633°, and the first obstacle is positioned at a

**Table 1. Experimental conditions table.**

|  | Initial Position | Pivot Position (cm) | Pendulum Length (cm) |
|---|---|---|---|
| Condition One: No Obstruction | 30° | None | 40cm |
| Condition Two: One Obstruction | 30° | 0° (0, -20) | 40cm; 20cm |
| Condition Three: Two Obstructions | 30° | 0°(0, -20); -30°(10, -34.641) | 40cm;20cm;10cm |

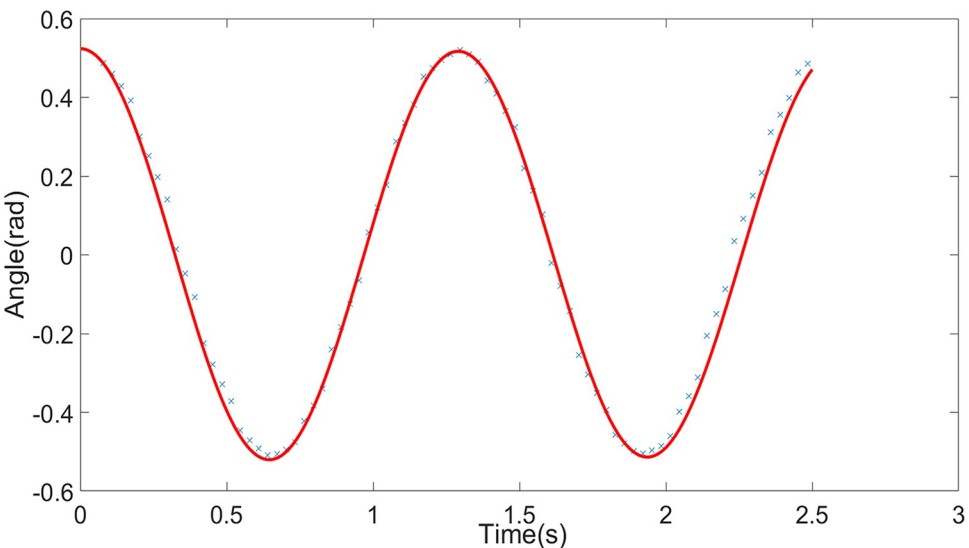

**Fig 5. Comparison between experimental and simulation data for condition pne (pendulum without obstruction).**

distance of 0.16 meters from the origin, while the second obstacle is placed at an angle of -20°
and a distance of 0.0462 meters from the origin. The double-obstacle pendulum system alter-
nates continuously among these three conditions. Another set of experiments involves a single
pendulum with a length of $l_1$ and an initial angle of 34.9633°, which remains unobstructed dur-
ing its motion.

From the Fig 8, it is apparent that the time-domain representation of the single pendulum
exhibits a harmonic-like waveform, which gradually diminishes in amplitude over time due to
energy dissipation. In contrast, the double-obstacle pendulum displays a distinct inflection
point in its angular displacement curve at $\theta(t) = 0$, $\theta(t) = 20°$, indicating a notable impact from

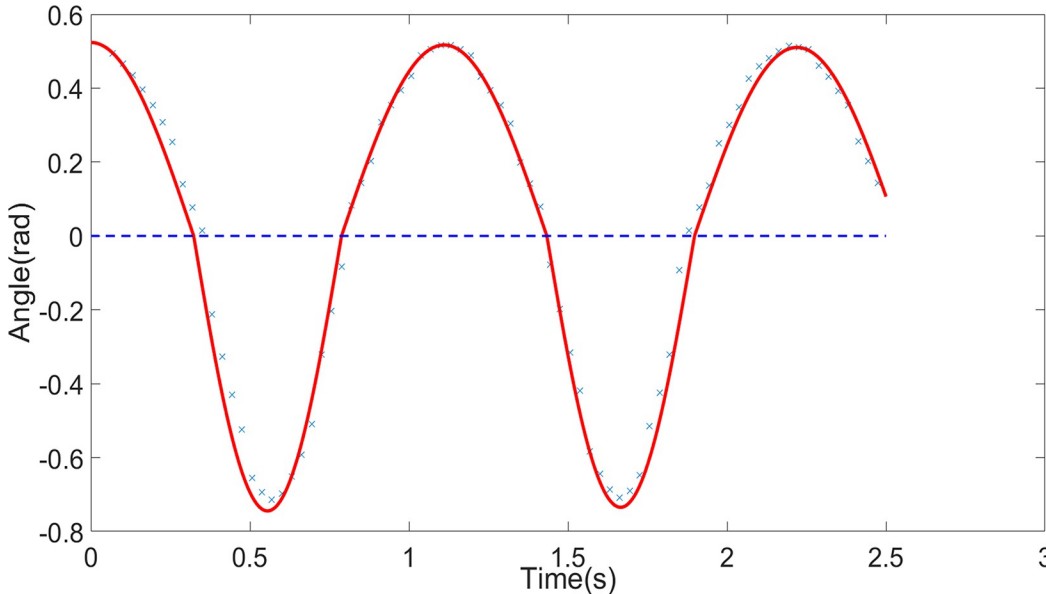

**Fig 6. Comparison between experimental and simulation data for condition two (pendulum with one obstruction).**

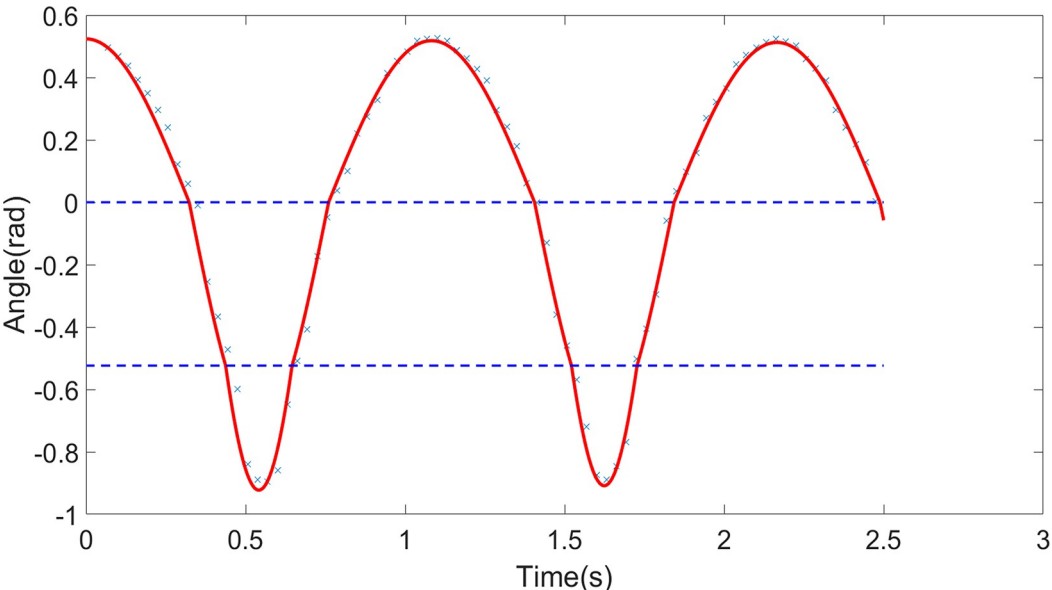

**Fig 7. Comparison between experimental and simulation data for condition three (pendulum with two obstructions).**

the obstacles. Specifically, after collisions with the obstacles, there is an increase in the amplitude and frequency of the motion of the single pendulum. This phenomenon is attributed to the significant alteration induced by the obstacles, resulting in a more pronounced amplitude and frequency response in the frequency spectrum analysis depicted in Fig 8B. These findings align with the anticipated outcomes, as the presence of obstacles in the double-pendulum system leads to a heightened amplitude and frequency response, thereby generating a richer spectrum of harmonic responses.

The investigation focuses on the influence of initial release angles on the dynamics of a double-obstacle pendulum system. The pendulum parameters and frequencies are detailed in Table 1, with obstacles positioned at 20˚ and 0˚ from the reference. Comparative simulations were conducted for initial release angles of 10˚, 20˚, and 30˚, as illustrated in Fig 9.

Observing the results, it becomes evident that as the initial angle increases, the influence of the two obstacles on the motion of the pendulum becomes more pronounced. When the pendulum makes contact with an obstacle, it undergoes a distinct change in its oscillatory behavior, resulting in variations in the operational conditions of the system. Larger initial angles result in higher angular velocities and amplitudes when the pendulum encounters the

**Table 2. System parameters.**

| Parameters | Values |
| --- | --- |
| $l_1(m)$ | 0.2396 |
| $l_2(m)$ | 0.0796 |
| $l_3(m)$ | 0.0334 |
| $\omega_1(Hz)$ | 6.3954 |
| $\omega_2(Hz)$ | 11.0957 |
| $\omega_3(Hz)$ | 17.1293 |
| $\xi$ | 0.01 |
| $\theta_0$ | 0.5273 |

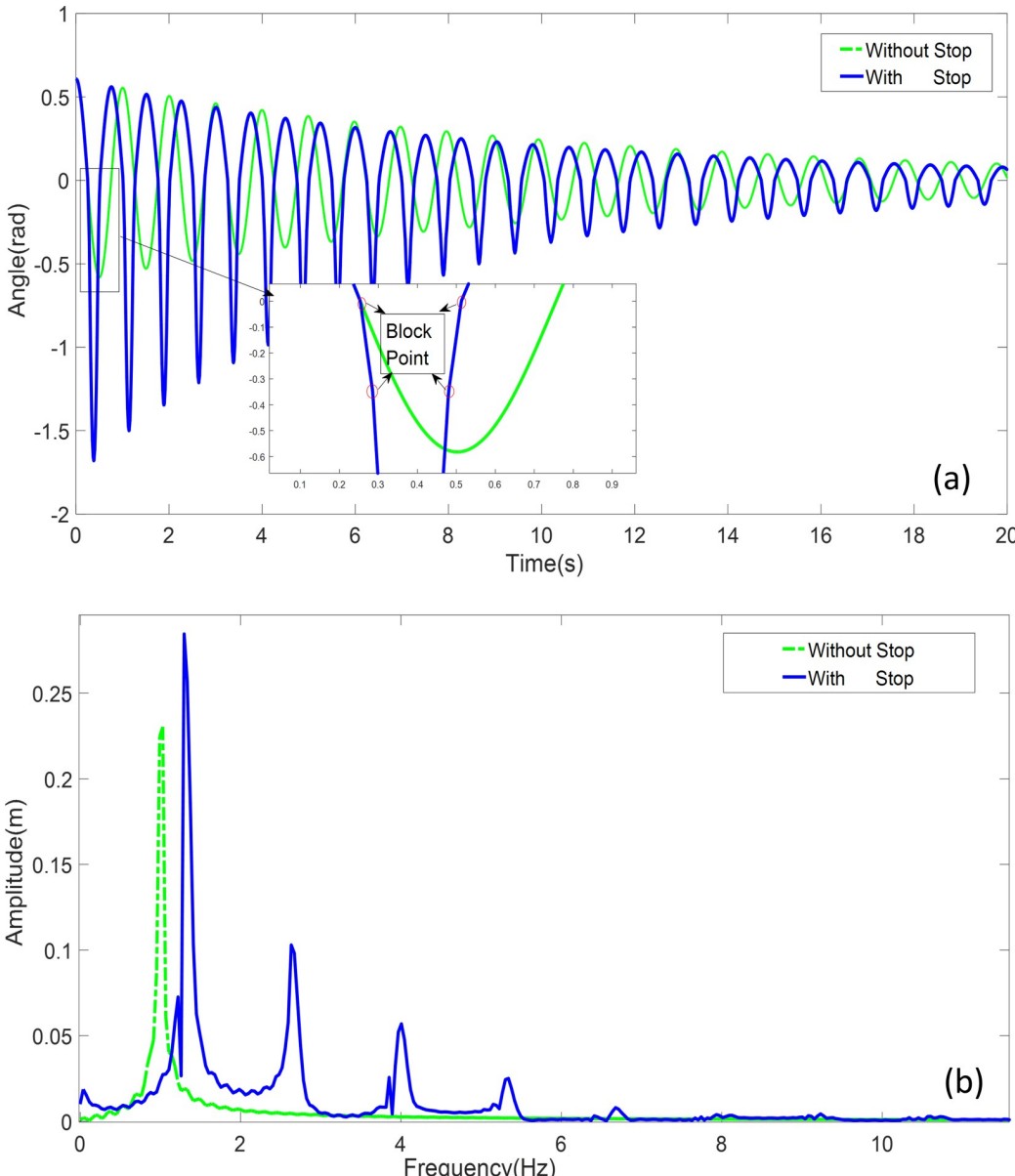

**Fig 8.** Unobstructed pendulum and double obstructed pendulum: **a** Time-domain simulation comparison **b** Frequency domain response analysis comparison.

obstacles. While there exists a minor distinction in frequencies, with slightly larger fundamental frequencies associated with larger initial angles, the influence of the obstacles on systems with larger initial angles is more conspicuous.

Fig 10A depicts a scenario in which the pendulum is released at an angle of 30˚, while the positions of the obstacles remain constant. In Fig 10B., the pendulum is released with an initial angle of 30˚ and a length of $l_1 = 0.6m$ $l_2 = 0.3m$ $l_3 = 0.1m$, while the angles and positions of both obstacles are altered. The purpose of Fig 6 is to investigate the impact of obstacles positioned at different locations on the oscillatory behavior of the double-obstacle pendulum system. As the pendulum interacts with obstacles at various positions, distinct sets of system

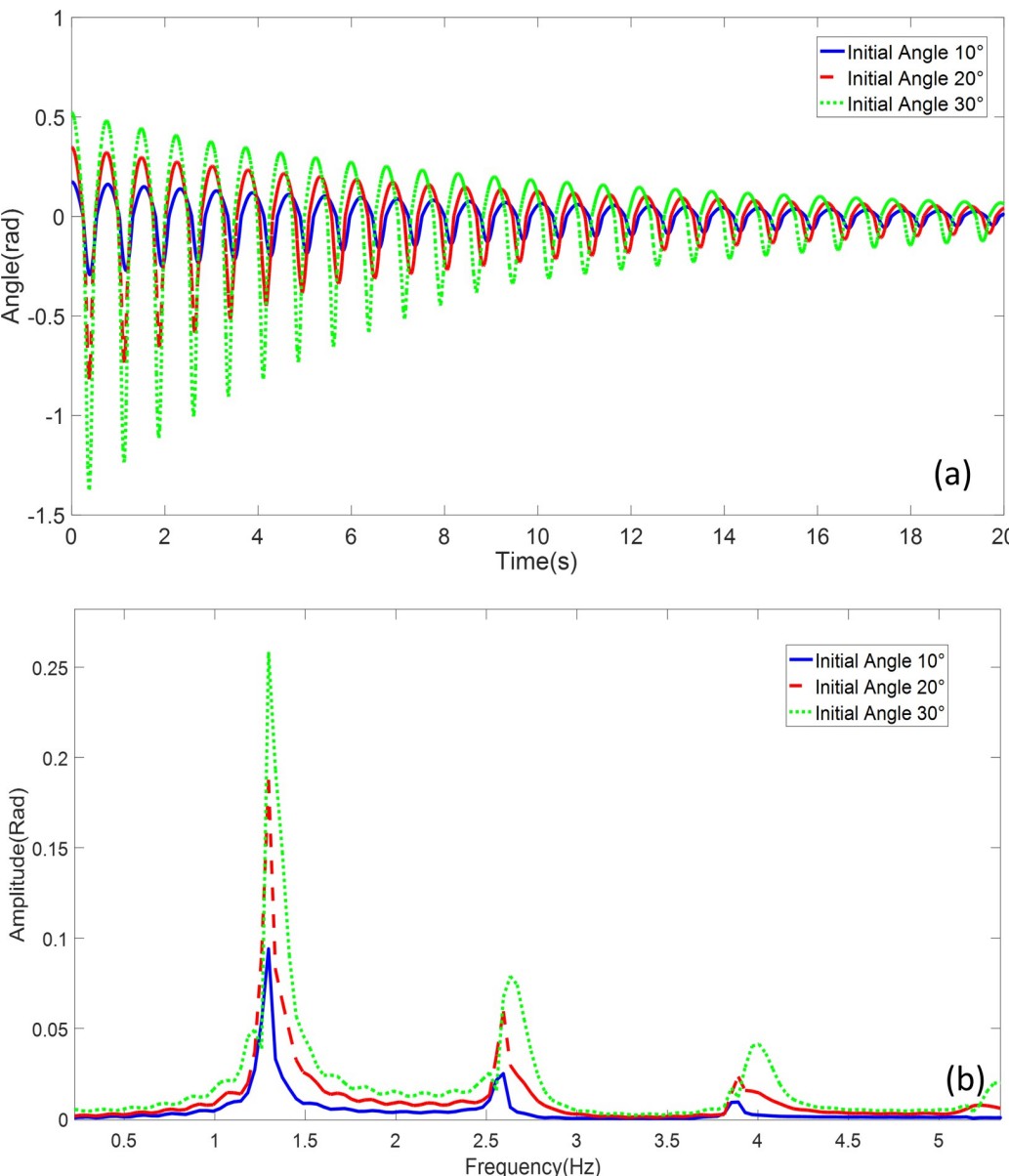

**Fig 9.** Double obstructed pendulum with different initial angles: **a** Time-domain simulation comparison **b** Frequency domain response analysis comparison.

parameters are generated, resulting in diverse frequencies and amplitudes in the motion of the system.

**Oscillation and phase plane analysis.** To further investigate the nonlinearity of the double-obstacle pendulum system, an in-depth nonlinear dynamic analysis was conducted on a double-obstacle single pendulum with constant pendulum length $l_1 = 0.6m$, $l_2 = 0.3m$, $l_3 = 0.15m$. The obstacles were situated at positions of 20˚ and -30˚, and the system was explored under varying initial angles and diverse damping conditions. The simulation encompassed the generation of time-domain diagrams and phase portraits to capture the oscillatory behavior of the double-obstacle pendulum system. The simulation duration extended to 20 seconds.

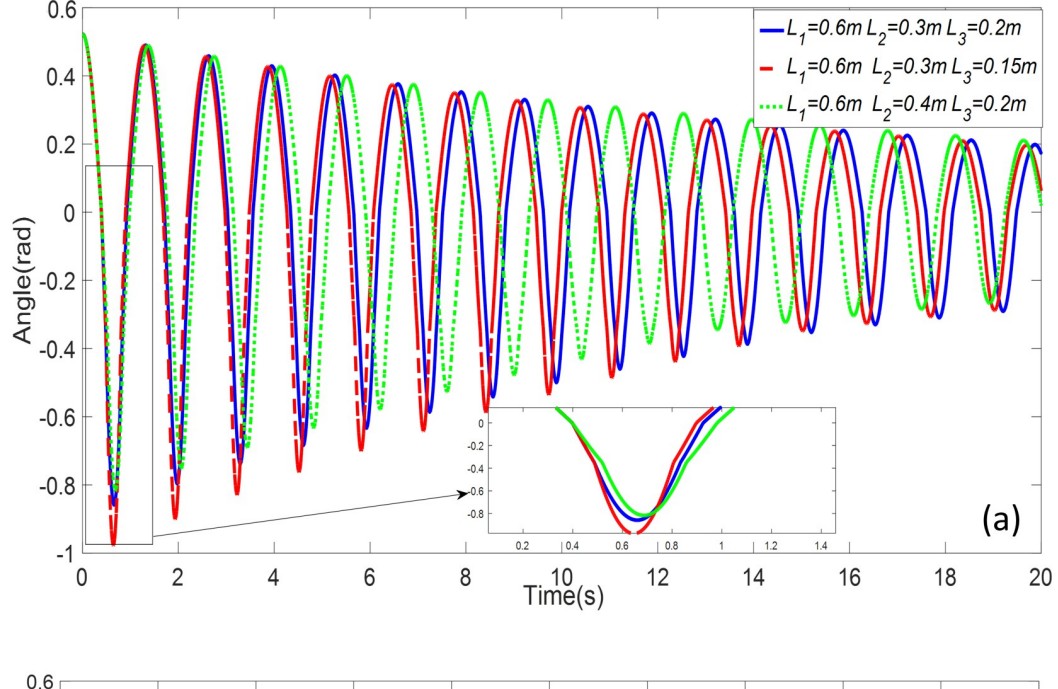

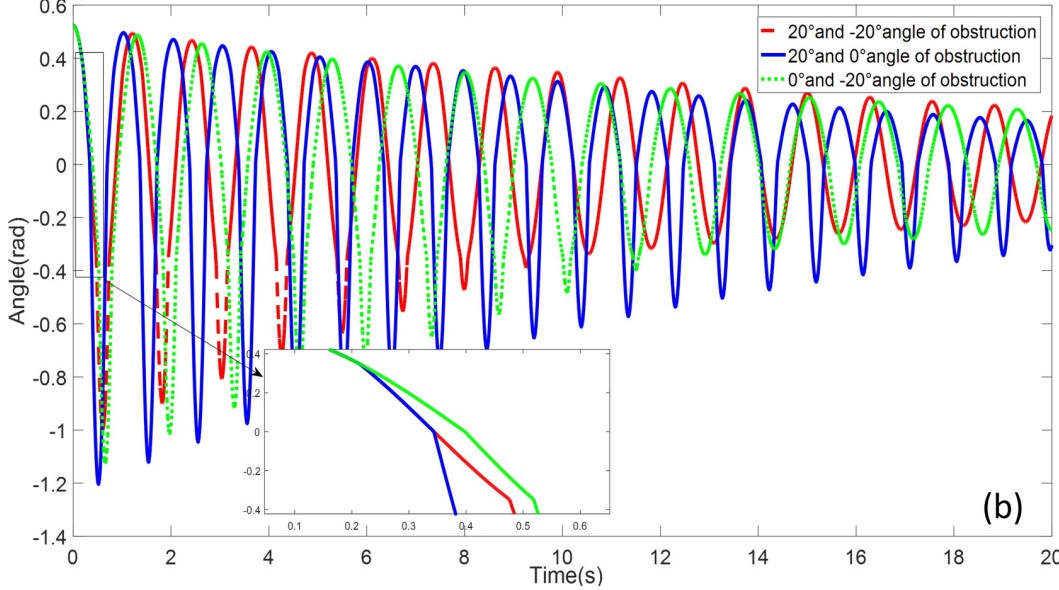

**Fig 10.** Double obstructed pendulum time-domain simulation: **a** Double obstructed pendulum with different pendulum lengths **b** Obstruction at different positions.

This analysis sought to examine the intricate interactions between initial conditions, damping effects, and the given obstacle positions. By systematically varying these parameters and employing high-fidelity numerical simulations, the study aimed to unravel the nonlinear dynamics that underlie the complex behavior of the double-obstacle pendulum system. The resulting time-domain diagrams and phase portraits provide a comprehensive view of the intricate motion patterns and evolution of the system over time.

Initially, the oscillations of a double obstructed pendulum released with an initial angle of 25˚ are examined, as depicted in Fig 11A and 11B. When the initial angle is relatively small,

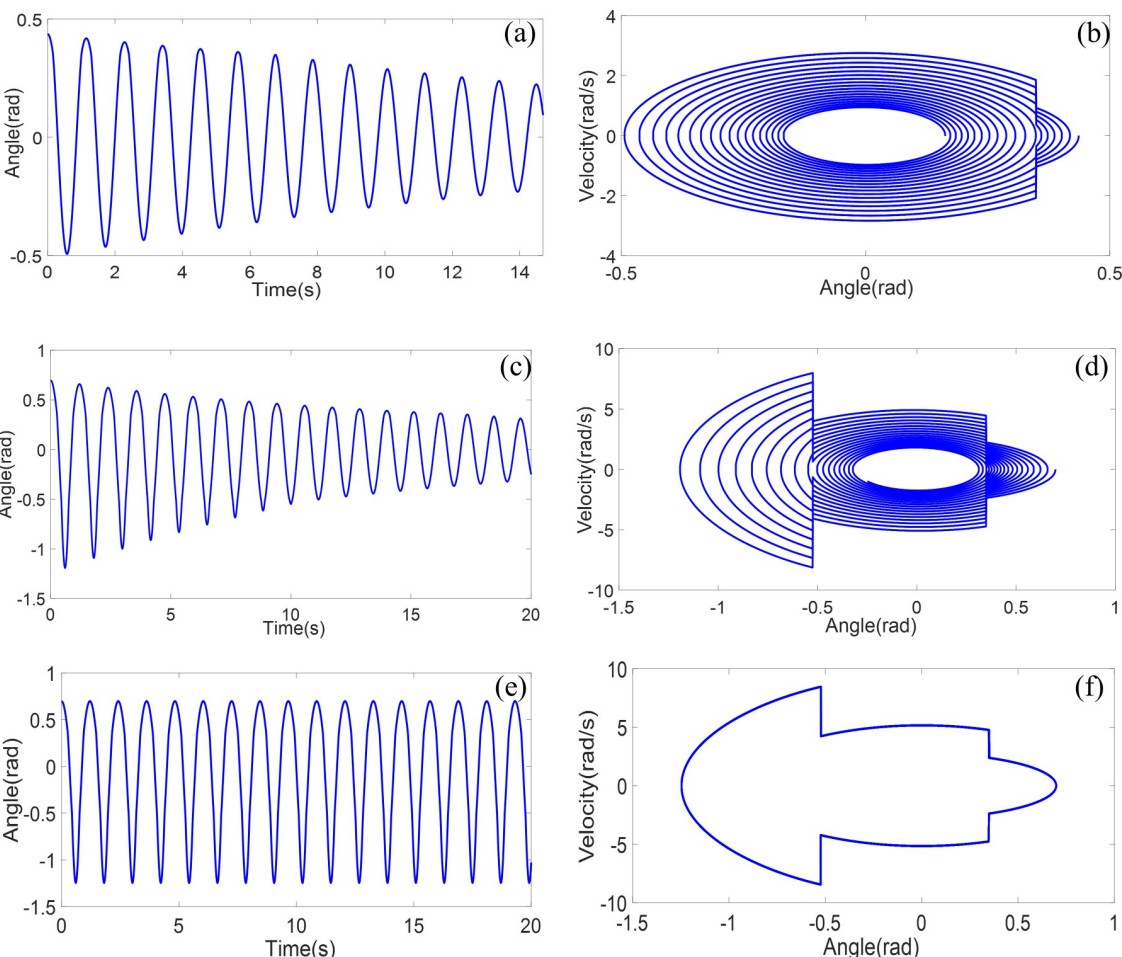

**Fig 11.** Double obstructed pendulum system time-domain simulation and phase diagram analysis: **a b** Initial angle 25° **c d** Initial angle 40° **e f** Initial angle 40° without damping.

the pendulum line does not collide with the second obstruction. As damping gradually dissipates energy, the amplitude diminishes. The corresponding phase portrait illustrates an abrupt change in velocity upon collision with the obstruction, consistent with the formula for velocity transformation during collisions.

Considering the matter of the initial release angle, and while maintaining constant pendulum length and obstruction positions, the initial angle is increased to 40°, as shown in Fig 11C and 11D. With a larger initial angle, the pendulum line is now able to contact the second obstruction. Consequently, this influences the angular velocity, frequency, and amplitude of the motion of the pendulum.

Subsequently, the impact of damping is examined by reducing the damping coefficient to zero, as depicted in Fig 11E and 11F. In this scenario, the double obstructed pendulum system does not experience energy dissipation, leading to unchanged amplitude and frequency.

Fig 11 provide insights into the diverse behaviors of the double obstructed pendulum system under these different conditions, shedding light on its nonlinear characteristics.

Considering the initial large angle of 80°, damping values of 0.01, 0.05, and 0.1, and a double-obstructed pendulum system, the simulation duration is 20 seconds, and the phase portrait is shown in Fig 12. In Figs 12D to 12A, as the damping gradually decreases, (d) with a damping

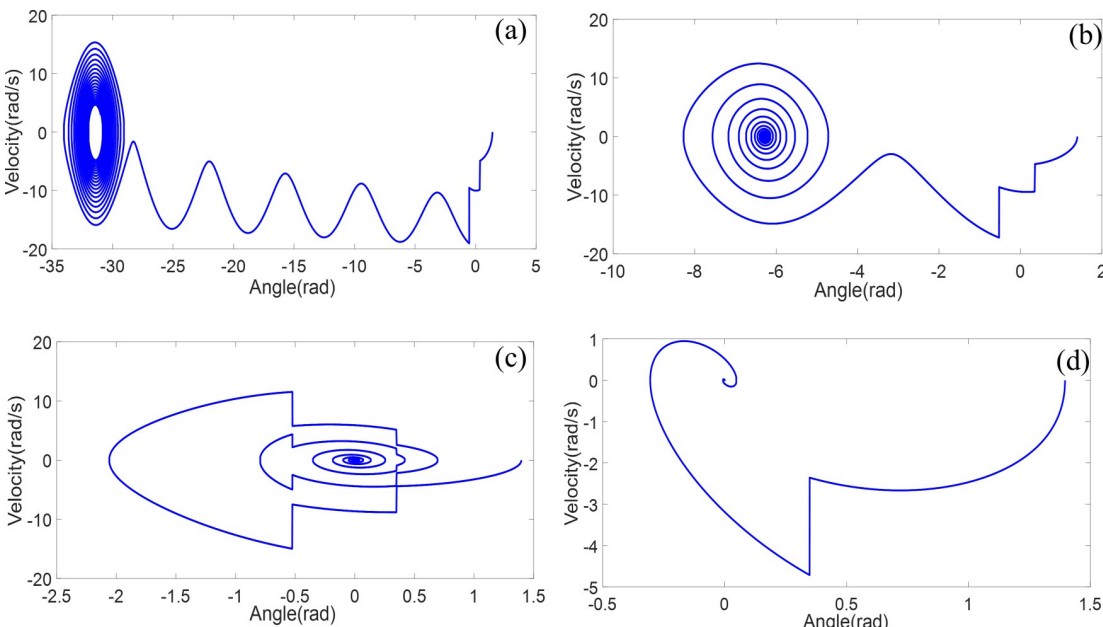

**Fig 12.** Phase diagram analysis of the double obstructed pendulum system with varying damping coefficients: **a** $\xi$ = **0.01 b** $\xi$ = **0.05 c** $\xi$ = **0.1 d** $\xi$ = **0.5**.

of 0.5 exhibits significant damping. At an angle of 30˚, the pendulum contacts the first obstruction, causing a sudden change in angular velocity. However, due to excessive damping, it does not reach the second obstruction, oscillating around the first obstruction until the velocity decreases to 0. (c) With a damping of 0.1, the pendulum continuously contacts both obstructions, and the velocity undergoes a sudden change at the obstruction positions. The angular displacement is always less than π, and the angular velocity changes its sign twice in each period. As energy is consumed, it fails to reach the second obstruction, oscillating around the first obstruction until the angular velocity becomes 0, halting the motion. (b) With a damping of 0.05, it rotates around the second obstruction twice. In the phase portrait, after passing the second obstruction at an angular displacement of -30˚, the angle increases, and the angular velocity becomes positive until energy decreases. It then oscillates around the second obstruction, with the velocity sign changing continuously until it becomes 0, stopping the motion. (a) With minimal damping and a large initial angle, after touching both obstructions, it continuously rotates around the second obstruction. The angular displacement keeps increasing, and the velocity remains negative. After rotating five times around the second obstruction, it oscillates around the second obstruction until the velocity becomes 0, stopping the vibration. The phase portrait indicates that when damping is minimal or absent, and the initial angle is large, the double-obstructed pendulum system exhibits rotational motion around the obstructions. These results demonstrate the significant impact of damping on the motion of the double obstructed pendulum.

## Conclusion

A novel approach, based on the principles of the RMTM, is proposed to investigate the motion of a single pendulum with abrupt changes in both large-angle release and pendulum length. This approach is applied to formulate a model for the single pendulum with sudden changes in both large-angle release and pendulum length. Numerical computations and experimental

measurements yield two sets of data, demonstrating a high degree of concordance between them. This validates the feasibility of the proposed model.

Employing high-precision numerical simulations, time-domain diagrams, frequency spectra, and phase portraits of a double-obstacle pendulum undergoing variations in pendulum length are obtained. The study explores the effects of obstacle positions and initial release angles on the motion pendulum. Notably, obstacles exert a substantial impact on the angular velocity of the pendulum motion. As the pendulum line contacts an obstacle, the angular velocity experiences an abrupt change, transitioning the pendulum into a different operational condition associated with reduced pendulum length. This change in length leads to a significant increase in frequency and amplitude. The initial angle has a distinct influence on the amplitude of the double-obstacle system; larger initial angles result in greater amplitudes. However, excessively large initial angles can cause the pendulum to rotate around the obstacles, leading to continuous increases in angular displacement. Different obstacle positions result in varying oscillation amplitudes and frequencies. Higher damping rates lead to more rapid energy dissipation within the double-obstacle pendulum system.

The novel approach of this paper toward modeling the obstacle-pendulum system, along with experimental image recognition and numerical simulations, opens a fresh avenue for exploring nonlinear behaviors in systems of pendulums.

## Supporting information

**S1 Fig. Experimental data for condition one (pendulum without obstruction).**
(TIF)

**S2 Fig. Experimental data for condition two (pendulum with one obstruction).**
(TIF)

**S3 Fig. Experimental data for condition three (pendulum with two obstructions).**
(TIF)

## Author Contributions

**Conceptualization:** Yang Yu, Hongtao Wei.

**Data curation:** Yang Yu, Xiangli Shi, Shouyu Cai.

**Funding acquisition:** Wei Wang.

**Methodology:** Jing Ma.

**Software:** Jiabin Wu.

**Writing – original draft:** Yang Yu.

**Writing – review & editing:** Jing Ma, Zilin Li, Wei Wang, Hongtao Wei, Ronghan Wei.

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
