## [Decision Letter · Decision Letter 0]

7 Dec 2023

PONE-D-23-34410Study on the Variable Length Simple Pendulum Oscillation Based on the Relative Mode Transfer MethodPLOS ONE

Dear Dr. Wei,

Thank you for submitting your manuscript to PLOS ONE. After careful consideration, we feel that it has merit but does not fully meet PLOS ONE’s publication criteria as it currently stands. Therefore, we invite you to submit a revised version of the manuscript that addresses the points raised during the review process.

ACADEMIC EDITOR: Major Revisions:

1- The authors disregard the elastic coefficient of the cable in their model, which limits the study when interruption due to obstacles is included. I recommend including the elastic coefficient in the model mainly due to the influence of the cable's restoring force;

2- Little has been discussed for cases where the pendulum's mass rotates around the obstacle. A more detailed study should be carried out about these problems;

3- Authors must justify why not use a triple pendulum with rigid links.

4- A justification for real applications must be included in the paper, and how the results presented can contribute;

5- Authors must include visual results of the pendulum movement;

6- More detailed comparisons of numerical and experimental results should be included. Highlighting what is numerical and what is experimental;

7- Experimental and numerical results for elastic cables must be included;

8- Numerical and experimental results for a triple pendulum with rigid links must be included.

Minor Revisions:

1- As a grammar correction, I would recommend, in the Introduction, you replace “Wright[11] consider” with “Wright[11] considered…”;

2- In Eq.(1), you could explain this equation. What is the relationship between the damping coefficient and the viscous friction?;

After Eq.(3), you could explain the difference between l and l ®.;

3- In Eq.(5), you may avoid misunderstanding if you change the order of the terms. I suggest you write as below. Otherwise, someone may think that θ(t_1 ) would be included in the sum. θ(t)=θ(t_i )+∑〖θ(t_j);

4- As a consequence of changes in Eq.(5), I would recommend to change the same in Eq.(6).

We look forward to receiving your revised manuscript.

Kind regards,

Angelo Marcelo Tusset

Academic Editor

PLOS ONE

Journal Requirements:

3. Thank you for stating the following in the Acknowledgments Section of your manuscript: "The study was supported by Key Scientific Research Projects of Universities in 

Henan Province (Grant No. 21A130003), Songshan Laboratory Project (Grant No.221100211000-01),

National Natural Science Foundation of China (Grant No. 12202400)."

Please remove any funding-related text from the manuscript and let us know how you would like to update your Funding Statement. Currently, your Funding Statement reads as follows: "The authors received no specific funding for this work."

Additional Editor Comments:

In this paper, the authors propose the application of the principle of the Relative Mode Transfer Method (RMTM) to establish a model for a single pendulum subject to sudden changes in its length. Numerical and experimental results are presented.

The proposal is interesting, and the paper can be considered for publication after a series of corrections.

Major Revisions:

1- The authors disregard the elastic coefficient of the cable in their model, which limits the study when interruption due to obstacles is included. I recommend including the elastic coefficient in the model mainly due to the influence of the cable's restoring force;

2- Little has been discussed for cases where the pendulum's mass rotates around the obstacle. A more detailed study should be carried out about these problems;

3- Authors must justify why not use a triple pendulum with rigid links.

4- A justification for real applications must be included in the paper, and how the results presented can contribute;

5- Authors must include visual results of the pendulum movement;

6- More detailed comparisons of numerical and experimental results should be included. Highlighting what is numerical and what is experimental;

7- Experimental and numerical results for elastic cables must be included;

8- Numerical and experimental results for a triple pendulum with rigid links must be included.

Minor Revisions:

1- As a grammar correction, I would recommend, in the Introduction, you replace “Wright[11] consider” with “Wright[11] considered…”;

2- In Eq.(1), you could explain this equation. What is the relationship between the damping coefficient and the viscous friction?;

After Eq.(3), you could explain the difference between l and l ®.;

3- In Eq.(5), you may avoid misunderstanding if you change the order of the terms. I suggest you write as below. Otherwise, someone may think that θ(t_1 ) would be included in the sum. θ(t)=θ(t_i )+∑〖θ(t_j);

4- As a consequence of changes in Eq.(5), I would recommend to change the same in Eq.(6).

Reviewers' comments:

Reviewer's Responses to Questions

**Comments to the Author**

1. Is the manuscript technically sound, and do the data support the conclusions?

Reviewer #1: Yes

Reviewer #2: Yes

2. Has the statistical analysis been performed appropriately and rigorously? 

Reviewer #1: I Don't Know

Reviewer #2: Yes

3. Have the authors made all data underlying the findings in their manuscript fully available?

Reviewer #1: Yes

Reviewer #2: Yes

4. Is the manuscript presented in an intelligible fashion and written in standard English?

Reviewer #1: Yes

Reviewer #2: Yes

5. Review Comments to the Author

Reviewer #1: I consider the paper has a good level and may be accepted after some improvements.

As a grammar correction I would recommend, in the Introduction, you replace “Wright[11] consider” by “Wright[11] considered…”

In the Eq.(1) you could demonstrate a explanation about this equation. What is the relationship between the damping coefficient and the viscous friction?

After the Eq.(3) you could explain the difference between l and l ®.

In Eq.(5) you may avoid misunderstanding if you change the order of the terms. I suggest you write as below. Otherwise, someone may think that θ(t_1 ) would be included in the sum.

θ(t)=θ(t_i )+∑〖θ(t_j)〗

As a consequence of changes in Eq.(5), I would recommend to change the same in Eq.(6).

Afterwards, the paper may be published.

Reviewer #2: The manuscript is interesting, however, it needs a major revision. It is not clear the novelty and contribution in Abstract and Conclusions. All figures need to be improved. The section Conclusion is confused.

6. PLOS authors have the option to publish the peer review history of their article (what does this mean?). If published, this will include your full peer review and any attached files.

Reviewer #1: No

Reviewer #2: No

---

## [Author Response · Author response to Decision Letter 0]

26 Jan 2024

Dear Editors and Reviewers,

Thank you for your careful review of our paper, "Study on the Variable Length Simple Pendulum Oscillation Based on the Relative Mode Transfer Method," and for providing valuable comments and suggestions. We greatly appreciate the diligent effort you have put into the review process.

In response to your comments, we have redesigned a more robust experimental plan, conducted multiple experiments, and engaged in detailed discussions and revisions. We have submitted the revised manuscript along with the "Revised Manuscript with Track Changes" to ensure the quality and content of the paper meet the high standards required. Below are our specific responses to each of your suggestions:

Reviewer #1:

1- As a grammar correction, I would recommend, in the Introduction, you replace “Wright[11] consider” with “Wright[11] considered…”;

Author’s response:

Sincere thanks for your evaluation of the grammar section of the paper. We have revised "Wright[11] consider" to "Wright[11] considered..." and polished the language throughout the entire paper. We have also thoroughly checked the grammar to ensure that the paper is grammatically correct, well-expressed, and maintains scientific rigor.

2- In Eq.(1), you could explain this equation. What is the relationship between the damping coefficient and the viscous friction?;

After Eq.(3), you could explain the difference between l and l ®.;

Author’s response:

Thank you very sincerely for your suggestions on the modeling equations of the paper. The damping coefficient in the context of mechanical systems, such as in the damped harmonic oscillator or the damped pendulum, is a measure of the strength of damping or resistance to motion in the system. It is associated with the dissipative forces that act against the motion of the system.

Viscous friction is a specific type of dissipative force that arises due to the motion of an object through a viscous medium, typically a fluid. The relationship between the damping coefficient and viscous friction depends on the specific context and the model being used. In this paper, the differential equation for the damped pendulum oscillation is derived from the fundamental principles of classical mechanics and vibration theory.

The differential equation for the large-angle pendulum with damping can be described as follows:

 represents the damping coefficient of the pendulum, indicating the magnitude of the damping.

We have provided explanations for Eq.(3)， represents the length and angular displacement of the pendulum when there is no obstruction. represents the length and angular displacement of the pendulum when obstructed. When obstructed, the angular displacement is measured from the obstruction point, with the left side considered positive and the right side considered negative. represents the derivative with respect to time.

3- In Eq.(5), you may avoid misunderstanding if you change the order of the terms. I suggest you write as below. Otherwise, someone may think that θ(t_1 ) would be included in the sum. θ(t)=θ(t_i )+∑〖θ(t_j);

Author’s response:

 Thank you very sincerely for your suggestions on the modeling equations of the paper. We have adjusted the sequence of Eq. (5): .

4- As a consequence of changes in Eq.(5), I would recommend to change the same in Eq.(6).

Author’s response: 

Thank you very sincerely for your suggestions on the modeling equations of the paper. We have adjusted the sequence of Eq. (6), ,and have reviewed all the formulas in the paper to ensure their correctness.

Reviewer #2:

1-The authors disregard the elastic coefficient of the cable in their model, which limits the study when interruption due to obstacles is included. I recommend including the elastic coefficient in the model mainly due to the influence of the cable's restoring force;

Author’s response:

 Thank you very sincerely for your suggestions on the paper. After careful consideration, we have made modifications and provided explanations before Eq.1 in the text regarding this issue. The pendulum is an idealized physical model, independent of the elasticity coefficient of the string. It consists of an idealized pendulum ball and string. The string, assumed to have an infinitely large elasticity coefficient, is massless and inelastic. The pendulum ball, with a significantly smaller radius than the length of the string, is considered a point mass. After encountering an obstruction, the elasticity coefficient of the string does not affect the pendulum's motion. The restoring force of the pendulum comes from the gravitational force in the downward direction, so the elasticity coefficient is not considered. In subsequent studies on the nonlinear vibration of beams and collisions, the influence of the elasticity coefficient will be taken into account.

2- Little has been discussed for cases where the pendulum's mass rotates around the obstacle. A more detailed study should be carried out about these problems;

Author’s response: 

Thank you very sincerely for your suggestions on the case study discussion of the paper. We have further modified and provided explanations for the rotational motion conditions of the pendulum around obstacles. The modifications are as follows: （Considering the initial large angle of 80°, damping values of 0.01, 0.05, and 0.1, and a double-obstructed pendulum system, the simulation duration is 20 seconds, and the phase portrait is shown in Fig 12. In Fig s 12d -12a, as the damping gradually decreases, (d) with a damping of 0.5 exhibits significant damping. At an angle of 30°, the pendulum contacts the first obstruction, causing a sudden change in angular velocity. However, due to excessive damping, it does not reach the second obstruction, oscillating around the first obstruction until the velocity decreases to 0. (c) With a damping of 0.1, the pendulum continuously contacts both obstructions, and the velocity undergoes a sudden change at the obstruction positions. The angular displacement is always less than π, and the angular velocity changes its sign twice in each period. As energy is consumed, it fails to reach the second obstruction, oscillating around the first obstruction until the angular velocity becomes 0, halting the motion. (b) With a damping of 0.05, it rotates around the second obstruction twice. In the phase portrait, after passing the second obstruction at an angular displacement of -30°, the angle increases, and the angular velocity becomes positive until energy decreases. It then oscillates around the second obstruction, with the velocity sign changing continuously until it becomes 0, stopping the motion. (a) With minimal damping and a large initial angle, after touching both obstructions, it continuously rotates around the second obstruction. The angular displacement keeps increasing, and the velocity remains negative. After rotating five times around the second obstruction, it oscillates around the second obstruction until the velocity becomes 0, stopping the vibration. The phase portrait indicates that when damping is minimal or absent, and the initial angle is large, the double-obstructed pendulum system exhibits rotational motion around the obstructions. These results demonstrate the significant impact of damping on the motion of the double obstructed pendulum.）

3- Authors must justify why not use a triple pendulum with rigid links.

Author’s response: 

We are very grateful to your comments on the manuscript. There has been extensive research on the triple pendulum with rigid links in the literature. After reviewing relevant papers, we selected one titled "Bifurcation and Chaos of Multi-body Dynamical Systems: Triple pendulum with rigid links," mainly applied in the study of bifurcation dynamics. However, it differs from the focus of our paper, which addresses nonlinear dynamics with a particular emphasis on the application of the Relative Mode Transfer Method (RMTM). Our study specifically addresses the nonlinear dynamics of large-angle damped pendulum, including the handling of nonlinear boundary changes when the pendulum string encounters obstacles. Subsequently, we will conduct further research on variable mass beams and variable length beams.

4- A justification for real applications must be included in the paper, and how the results presented can contribute;

Author’s response: 

We are very grateful to your comment on the manuscript. The vibration problem of a variable pendulum is a classical issue. In various important fields of life, mathematical models of variable-length single pendulum vibrations are needed. Examples include the swinging motion of a suspension cable when lifting cargo with a crane, the seismic isolation system for bridge foundations, and the control of horizontal vibrations in tall buildings. This paper establishes a model for a single pendulum system with abrupt changes in length based on the relative mode transfer method. The design involves comparing simulations with experimental validations to demonstrate the accuracy of the method. Numerical examples of a double-impeding single pendulum system with different parameters are calculated, and results include time-domain responses, frequency spectrum analysis, and phase diagram analysis. The model accurately reflects the variable-length pendulum system, providing a new approach for studying similar nonlinear problems. From the contact of the pendulum line with one obstacle to two obstacles, it can be extrapolated to an infinite number of obstacles, offering preliminary research for studying momentary variations in pendulum length, variable-mass beams, and variable-length beams. This research can be applied to modeling and solving vibrations occurring in the extension and retraction of pendulum arms, such as those in cranes. It also provides a modeling approach for vibrations during the extension of antennas in aerospace satellites.

5- Authors must include visual results of the pendulum movement;

Author’s response: 

We are very grateful to your comments on the manuscript. We have adjusted the experimental design for the large-angle pendulum vibration, where the pendulum encounters one or two obstacles. A new experimental platform was set up, employing a more accurate design. Multiple experiments were conducted for the large-angle pendulum without obstacles, with one obstacle, and with two obstacles. The experimental results were compared with numerical simulation results, and the two sets of data closely matched within an acceptable range of error. Visual results of the pendulum experiments are presented in the paper, as shown in Fig 3 and 4, effectively reflecting the visual outcomes. Additionally, we have uploaded relevant experimental videos in the appendix, providing an intuitive representation of the motion of the large-angle damped pendulum and its behavior upon encountering obstacles.

6- More detailed comparisons of numerical and experimental results should be included. Highlighting what is numerical and what is experimental;

Author’s response: 

We sincerely appreciate your evaluation of the paper; these comments are valuable for improving our article. We have made extensive revisions to our manuscript and added extra data, enhancing the persuasiveness of our results. In the revised experimental study presented in the paper, we have re-written the section discussing the experiments and numerical simulations. Detailed descriptions of the simulation methods, step sizes, and simulation conditions were provided. Multiple experiments were conducted for three different conditions. A comprehensive comparison between simulation results and numerical simulation results was included, as shown in Figures 5 to 7. This enriches the experimental content of the paper and provides clearer insights.

7- Experimental and numerical results for elastic cables must be included;

Author’s response: 

Thank you for your suggestions. We have re-evaluated the impact of the elasticity coefficient on the large-angle pendulum. We provided detailed numerical and experimental data and results for the large-angle pendulum touching obstacles. The results show that the simulation can accurately predict the motion of the large-angle damping pendulum with one or two obstacles. This indicates that the model established through the Relative Mode Transfer Method (RMTM) can accurately solve the motion laws of the large-angle pendulum with variable length. The pendulum is an ideal physical model, independent of the elasticity coefficient of the string. It consists of an idealized pendulum ball and string. The string is assumed to be massless and inelastic, with an idealized infinite elasticity coefficient. When the pendulum touches an obstacle, the obstacle is treated as a rigid body without deformation.

8- Numerical and experimental results for a triple pendulum with rigid links must be included.

Author’s response: 

Thank you for your suggestions. We have consulted relevant literature and thoroughly studied the triple pendulum with rigid links, mainly applied to the study of bifurcation dynamics. However, this differs from the focus of our paper, which addresses nonlinear dynamics, applying the Relative Mode Transfer Method (RMTM) to handle the nonlinear boundary changes when the pendulum string touches obstacles during large-angle damping pendulum motion. In the future, we plan to further study variable mass beams and beams with variable lengths. We have redesigned and conducted multiple experiments, adding new conditions to validate the reliability of the method. The paper has been extensively revised to include a significant amount of numerical and experimental data.

---

## [Decision Letter · Decision Letter 1]

9 Feb 2024

Study on the Variable Length Simple Pendulum Oscillation Based on the Relative Mode Transfer Method

PONE-D-23-34410R1

Dear Dr. Wei,

We’re pleased to inform you that your manuscript has been judged scientifically suitable for publication and will be formally accepted for publication once it meets all outstanding technical requirements.

Kind regards,

Angelo Marcelo Tusset

Academic Editor

PLOS ONE

Additional Editor Comments (optional):

The authors presented a fully revised version, meeting all the requested corrections and the criteria required for publication of this Journal.

After these considerations, I consider the paper accepted in its current form.

Reviewers' comments:

Reviewer's Responses to Questions

**Comments to the Author**

1. If the authors have adequately addressed your comments raised in a previous round of review and you feel that this manuscript is now acceptable for publication, you may indicate that here to bypass the “Comments to the Author” section, enter your conflict of interest statement in the “Confidential to Editor” section, and submit your "Accept" recommendation.

Reviewer #1: All comments have been addressed

Reviewer #2: All comments have been addressed

2. Is the manuscript technically sound, and do the data support the conclusions?

Reviewer #1: Yes

Reviewer #2: Yes

3. Has the statistical analysis been performed appropriately and rigorously? 

Reviewer #1: I Don't Know

Reviewer #2: Yes

4. Have the authors made all data underlying the findings in their manuscript fully available?

Reviewer #1: Yes

Reviewer #2: Yes

5. Is the manuscript presented in an intelligible fashion and written in standard English?

Reviewer #1: Yes

Reviewer #2: Yes

6. Review Comments to the Author

Reviewer #1: (No Response)

Reviewer #2: The authors have addressed all comments. However, the authors have improved the quality of the figures.

7. PLOS authors have the option to publish the peer review history of their article (what does this mean?). If published, this will include your full peer review and any attached files.

Reviewer #1: **Yes: **Rafael Henrique Avanço

Reviewer #2: **Yes: **Antonio Marcos Batista

---

## [Editor Report · Acceptance letter]

27 Mar 2024

PONE-D-23-34410R1 

PLOS ONE

Dear Dr. Wei, 

I'm pleased to inform you that your manuscript has been deemed suitable for publication in PLOS ONE. Congratulations! Your manuscript is now being handed over to our production team.

Kind regards, 

on behalf of

Professor Angelo Marcelo Tusset 

Academic Editor

PLOS ONE